# Multitrophic and Multilevel Interactions Mediated by Volatile Organic Compounds

**DOI:** 10.3390/insects15080572

**Published:** 2024-07-28

**Authors:** Dongsheng Niu, Linbo Xu, Kejian Lin

**Affiliations:** 1Institute of Grassland Research, Chinese Academy of Agricultural Sciences, Hohhot 010000, China; niudongsheng@caas.cn; 2Inner Mongolia-CABI Joint Laboratory for Grassland Protection and Sustainable Utilization, Hohhot 010000, China; 3Key Laboratory of Biohazard Monitoring, Green Prevention and Control for Artificial Grassland, Ministry of Agriculture and Rural Affairs, Hohhot 010000, China; 4Inner Mongolia Key Laboratory of Grassland Protection Ecology, Hohhot 010000, China

**Keywords:** volatile organic compounds, multitrophic interactions, multilevel interactions, herbivore-induced plant volatiles, green leaf volatiles, aboveground–belowground interactions, chemical defense mechanisms

## Abstract

**Simple Summary:**

Plants and their insect neighbors engage in a silent dialogue that is vital for ecological health, a phenomenon often invisible to the naked eye. Scientists have unveiled a fascinating aspect of this dialogue: plants communicate using a unique language of scents known as volatile organic compounds (VOCs). These aromatic messages serve as a defense system, recruiting allies in the form of beneficial insects or broadcasting alerts to fellow flora about impending threats such as pests. Our investigation delves into the mechanics of these aromatic signals across diverse habitats, from the dense canopies of forests to the cultivated expanses of farmlands. We discovered that these VOCs are not just background noise but are instrumental in maintaining ecological harmony. They hold the potential to revolutionize our approach to crop protection, offering eco-friendly alternatives to conventional pesticides. This research is groundbreaking, as it deciphers nature’s olfactory code, paving the way for harnessing VOCs to foster environmental sustainability and food security. By mimicking the botanical strategies observed in plants, we can devise sustainable solutions to agricultural challenges, steering us towards a greener and more resilient future.

**Abstract:**

Plants communicate with insects and other organisms through the release of volatile organic compounds (VOCs). Using Boolean operators, we retrieved 1093 articles from the Web of Science and Scopus databases, selecting 406 for detailed analysis, with approximately 50% focusing on herbivore-induced plant volatiles (HIPVs). This review examines the roles of VOCs in direct and indirect plant defense mechanisms and their influence on complex communication networks within ecosystems. Our research reveals significant functions of VOCs in four principal areas: activating insect antennae, attracting adult insects, attracting female insects, and attracting natural enemies. Terpenoids like α-pinene and β-myrcene significantly alter pest behavior by attracting natural enemies. β-ocimene and β-caryophyllene are crucial in regulating aboveground and belowground interactions. We emphasize the potential applications of VOCs in agriculture for developing novel pest control strategies and enhancing crop resilience. Additionally, we identify research gaps and propose new directions, stressing the importance of comparative studies across ecosystems and long-term observational research to better understand VOCs dynamics. In conclusion, we provide insights into the multifunctionality of VOCs in natural ecosystems, their potential for future research and applications, and their role in advancing sustainable agricultural and ecological practices, contributing to a deeper understanding of their mechanisms and ecological functions.

## 1. Introduction

Volatile organic compounds (VOCs) are increasingly recognized for their pivotal roles in plant ecology and agricultural research [1,2]. These compounds are not only integral to plant communication but also play a crucial role in the interactions between plants and a diverse array of organisms [3], including insects [4], fungi [5,6], bacteria [7,8], and viruses [9,10]. The variety and complexity of VOCs render them essential subjects for exploring plant defense mechanisms and the dynamics of ecosystems [11,12].

In natural settings, plants emit a multitude of VOCs, the types and concentrations of which fluctuate in response to external environmental [13] and biotic pressures [14]. Through these volatile substances, plants engage in communication with their surrounding biotic community, triggering both direct and indirect defense responses [15]. These chemical signals, emitted by plants infested by herbivores, are capable of attracting natural enemies to mitigate herbivorous insect damage [16] and can also bolster the defense capabilities of neighboring plants [15,17], thereby contributing to the balance within ecosystems [18].

This review is dedicated to examining the role of VOCs in multitrophic interactions, especially their dual function in direct and indirect plant defense mechanisms. We delve into the function of VOCs in the chemical communication between plants, insects, and antagonists and how they influence the dynamic equilibrium within ecosystems. Additionally, we discuss the potential applications of VOCs in agricultural production and their significance in future ecological research [3].

Through a systematic review of the existing literature and an analysis of the latest studies, we aim to provide readers with a comprehensive perspective on the multifunctionality of VOCs within ecosystems and their potential in future research and applications. As we deepen our understanding of the mechanisms and ecological functions of VOCs, we are confident that they will emerge as new tools for protecting and enhancing our precious ecosystems, bridging knowledge gaps, and defining research objectives.

## 2. Methods

### 2.1. Comprehensive Literature Retrieval Strategy

A comprehensive systematic search was conducted on 10 May 2024, utilizing the Scopus and Web of Science databases. Relevant publications were retrieved from the Web of Science Core Collection (WOSCC) and Scopus in BibTeX format, employing a stepwise retrieval method and its combination, as detailed in Table 1 [19]. No time-period limit was imposed, allowing for the inclusion of all scientific literature within the study domain. For the comprehensive literature review on VOCs regulating multitrophic and multilevel interactions, we adhered to the PRISMA (Preferred Reporting Items for Systematic Reviews and Meta-Analyses) guidelines (Figure 1) [20]. Only journal publications in English were included, while conference proceedings, books, and book chapters were excluded [21].

### 2.2. Criteria for Selecting Literature

A sample size exceeding 200 research documents was considered adequate for a robust analysis [22]. Manual screening was conducted using Notepad++ version 8.6.4 to meticulously evaluate and focus on full research papers pertinent to the topic, ensuring the selection of the most relevant documents. Studies not related to the research theme were excluded based on abstract and title assessments, with validation from three authors to confirm their relevance. Efforts were directed towards identifying and rectifying errors in affiliation data and standardizing the names of authors and institutions. Ultimately, the chosen documents (406 papers) were archived as BibTeX files for further analysis.

## 3. Sources and Classification of VOCs

### 3.1. An Overview of Plant-Derived VOCs

Plants, as sessile organisms, continually adapt to their environments to ensure survival and reproductive success. Central to this adaptive strategy is their production of specialized metabolites, known as plant secondary metabolites. Among these, VOCs are characterized by a low molecular weight (less than 250) and high vapor pressure (boiling point below 340 °C) at ambient temperatures (0.01 KPa, 20 °C) [23]. These physical properties facilitate their diffusion across cellular membranes and release into the surrounding environment [24]. VOCs are synthesized across diverse plant organs, encompassing roots, stems, leaves, fruits, seeds, and, notably, flowers, which serve as the most prolific reservoirs in terms of both abundance and variety of VOCs [25,26,27]. Researchers have identified over 200,000 specialized metabolites [28], among which approximately more than 1700 VOCs have been documented across 90 distinct angiosperm and gymnosperm families [27].

### 3.2. Biosynthetic Pathways and Categories of VOCs

Throughout their life cycles, plants emit an array of VOCs [29], including alcohols, aldehydes, acids, esters, ketones, and terpenes [27]. These compounds combine to form a species-specific “olfactory fingerprint” that is unique to each plant species [30]. This olfactory signature is categorized into two types: general odorants, prevalent among various plants, comprising elements such as straight-chain alcohols, aldehydes, and esters with six carbon atoms, unsaturated fatty acids, and terpenes; and specific odorants, exclusive to particular species, emanating from the degradation of secondary metabolites with phylogenetic relevance [31].

VOCs can be classified based on their biosynthetic origins into four main categories: terpenes [32], phenylpropanoids [33,34], fatty acid derivatives [35], and amino acid derivatives [36] (Figure 2). Terpenoids represent the most extensive and varied class of secondary metabolites found in nature [37], consisting of a range of structures derived from five-carbon isoprene units. This includes monoterpenes, sesquiterpenes, diterpenes, triterpenes, and tetraterpenes [38]. Notable examples are vincristine, β-caryophyllene, paclitaxel, soyasaponins, artemisinin, and carotenoids [39]. The second major group, benzenoid and phenylpropanoid compounds [36], are synthesized from aromatic amino acids like phenylalanine. The third category comprises fatty acids and their derivatives, which are synthesized in plant biosynthesis, predominantly consisting of C_18_ polyunsaturated fatty acids such as linoleic and linolenic acids [40]. Examples of this category include hexanal, cis-3-hexen-1-ol, nonanal, and methyl jasmonate. The fourth category includes compounds derived from amino acids, characterized by the presence of both amino and carboxyl groups.

Although they represent the smallest proportion of total volatiles, many VOCs, particularly those rich in floral volatiles (FVs) and fruity volatiles, originate from amino acids such as alanine, valine, and leucine, or their biosynthetic intermediates [41]. These compounds are sequestered within the plant in diverse manners. Terpenoids and benzenoid compounds tend to accumulate in high concentrations within the leaves, whereas water-soluble VOCs, along with hydrophobic monoterpenes and sesquiterpenes, are typically stored in smaller quantities and for shorter durations [42].

Research into the functionality of VOCs has delineated four principal areas of interest: compounds that activate insect antennae, attractants for adult insects, attractants for female adults, and attractants for natural enemies (Figure 2). Each category plays a unique biological role in plant adaptation and defense mechanisms. However, additional research is essential to fully elucidate the specific functions and implications of each class of volatiles.

## 4. The Role of VOCs in Multitrophic (Plant–Pest–Antagonist) Interactions

Under natural conditions, plants typically emit no more than about 30 types of VOCs. The composition of these plant volatiles is highly variable, reflecting the dynamic nature of plants as they respond to abiotic factors such as mechanical damage [43,44,45,46], plant age [47], temperature [48], drought [49,50,51], ozone (O_3_) [52], CO_2_ [50], and light conditions [53]. Biotic factors, including insects and microorganisms, can also alter the physiological state of plants, thereby influencing VOCs’ emissions. Moreover, variations in plant volatiles may occur among individuals within the same species, affected by factors such as genotype and cultivar [54].

In the face of threats from both biotic and abiotic stresses, plants have developed an array of resistance mechanisms over their evolutionary history [55]. While non-volatile secondary metabolites, including alkaloids and terpenes, play a pivotal role in direct defense responses, VOCs are instrumental in both direct and indirect defense strategies [56]. Upon encountering environmental stresses, VOCs sequestered within specialized structures can be swiftly released in response to alterations in vapor pressure [57,58]. Essential compounds such as monoterpenes, sesquiterpenes, stress-induced green leaf volatiles (GLVs), and other terpenes are integral to the activation of plant defense mechanisms [44].

### 4.1. Chemical Defense Mechanisms of Plants against Pests

VOCs are pivotal in plant ecosystems, serving as key mediators of inter-plant communication and significantly influencing the behavior and ecological interactions of various organisms associated with plants. Many herbivorous insects, particularly nocturnal species, depend on environmental background odors, host plant volatiles, and non-host plant volatiles for activities such as foraging, mating, and oviposition [59,60,61]. Significant progress has been made in understanding the olfactory cues and mechanisms by which insects locate their hosts, particularly in the orders Lepidoptera [62], Diptera [63], Coleoptera [64], and Hymenoptera [65].

Through coevolution with herbivores, plants have evolved diverse strategies to adapt to or mitigate herbivory [66]. In response to herbivore attacks, plants engage in self-defense by emitting toxic, deterrent, and anti-nutritional VOCs [67]. Signal-induced indirect defenses are initiated when herbivores prompt plants to release VOCs; the damaged tissues emit signals that propagate, are perceived by the same or neighboring plants, and activate corresponding defense responses [27,68].

Plant defense-induced volatile compounds include aldehydes, alcohols, esters, and alkanes. These are divided into two categories based on their properties. The first category comprises non-specific defensive volatiles, primarily induced by herbivorous arthropod feeding, such as the common fatty acid derivatives known as green leaf volatiles (GLVs) (Table 2). Normally, healthy plants emit minimal GLVs; their production is significantly triggered in response to herbivory or environmental stress (Table 2). The second category consists of specific defensive volatiles, predominantly volatile terpenes (including monoterpenes and sesquiterpenes hydrocarbons and their derivatives), nitrogen-containing compounds, methyl salicylate, and indole [27,69,70].

### 4.2. Chemical Communication between Pests and Antagonists

When plants are attacked by insects, they release herbivore-induced plant volatiles (HIPVs), which play a crucial role in plant defense by protecting nearby undamaged plants [71]. HIPVs primarily include shikimate pathway products, fatty acid derivatives, and terpenes. By releasing HIPVs, plants can enhance their defense responses and modulate insect behavior to counteract herbivorous insect damage (Figure 3).

Herbivorous insect activity on plants triggers not only the passive dispersal of GLVs and pre-stored volatile compounds but also prompts the active biosynthesis and emission of new volatiles, including terpenes, aldehydes, alcohols, ketones, esters, indoles, and certain furan derivatives (Table 2). Typically, herbivore-attacked plants exhibit a marked increase in the diversity and quantity of terpenoid compounds relative to their healthy counterparts. Distinct variations in the HIPVs emitted can be observed when the same plant species is assailed by different herbivorous insects or when various plant species are targeted by the same insect (Table 2). HIPVs elicit defense reactions in neighboring plants, covering both direct and indirect defensive strategies. Direct defense entails the reception of HIPV signals by adjacent plants, leading to the activation of defense genes, modulation of plant hormones, and synthesis of defense-related secondary metabolites, thus fortifying physiological resistance to herbivorous insects (Figure 3) [72,73]. Indirect defense involves HIPV-stimulated plants secreting extrafloral nectars (EFNs) or emitting volatiles that lure predatory or parasitic insects, thereby indirectly diminishing herbivore predation [74]. Moreover, HIPVs serve not only to initiate defense responses in the afflicted plant tissues but also to prime adjacent tissues for potential herbivory. Tissues pre-exposed to HIPVs promptly engage defense mechanisms upon herbivore incursion, effectively curtailing herbivore damage (Figure 3) [75,76]. Beyond bolstering plant defenses against herbivorous insects, certain HIPVs also function as defense suppressors, impeding the plant’s defensive capabilities against herbivores [77].

### 4.3. Importance of VOCs in Plant Defense against Infection

The release of VOCs by plants plays a crucial role in their defense strategy against infections and herbivores. VOCs serve as chemical signals that can attract natural enemies of herbivores, such as parasitoids and predators [78,79], thereby reducing herbivory and enhancing plant survival [80]. Recent studies have highlighted the multifunctionality of VOCs in ecological interactions. For example, limonene emitted by citrus plants not only repels herbivores but also attracts natural enemies and pollinators, showcasing its ecological pleiotropy [80,81]. Moreover, VOCs can also induce resistance in neighboring plants, preparing them for potential attacks by priming their defensive pathways [16]. These multifaceted roles of VOCs not only help individual plants combat biotic stress but also contribute to the overall resilience and balance of ecosystems.

However, the screening and identification of key volatile organic compounds (VOCs) is a systematic process that typically involves several steps: volatile sampling, volatile screening and identification, and behavioral validation (Appendix A). In brief, variations in plant volatiles before and after herbivory are usually analyzed using gas chromatography-mass spectrometry (GC-MS). Newly detected compounds must be assessed using gas chromatography–electroantennographic detection (GC-EAD) to determine if they elicit antennal responses in insects. Further electroantennogram (EAG) measurements are conducted to establish the concentration at which the maximum response occurs, providing the basis for subsequent behavioral validation and field experiments.

**Table 2 insects-15-00572-t002:** Major compounds of GVs, GLVs, and HIPVs.

Type	Major Compound	Biological Function	Plant	Insect	Reference
GVs	Hexyl acetate	Female adult repellents		*Cydia pomonella*	[82]
	Pentadecane, Nonanal, α-farnesene	Antennal active substances	*Grapevine*	*Lobesia botrana*	[83]
	2-phenyl ethanol, Methyl salicylate, Decanal	Adult attractants	*Malus domestica*	*Argyresthia conjugella*	[84]
	4-hydroxy-4-methyl-2-pentanone, 1-hepten-3-one, 1-octen-3-ol, 3-octanone	Oviposition attractants	*Phaseolus vulgaris*	*Delia platura*	[85]
	(Z)-3-hexenol, (E)-2-hexenol, (E)-2-hexenal, Hexanal	Antennal active substances	-	*Episyrphus balteatus*	[86]
	3-octanol, 1-octen-3-ol	Antennal active substances	-	*Thanasimus formicarius*	[87]
	(E)-4,8-dimethylnona-1,3,7-triene, (E,E)-4,8,12-trimethyltrideca-1,3,7,11-tetraene, (E)-ocimene	Adult attractants	*Gossypium* spp.	*Anthonomus grandis*	[88]
	(E)-2-hexenyl acetate, (±)-Linalool, (E,E)-α-farnesene, (Z)-3-hexen-1-ol, (S)-(-)-α-pinene, Methyl salicylate	Adult repellents	Alfalfa	*Lygus hesperus*	[89]
	Methyl salicylate, Cis-hex-3-en-1-ol	Nature enemy attractants	*Solanum lycopersicum*	*Macrosiphum euphorbiae*, *Aphidius ervi*	[90]
	Nerol, α-campholenol, p-cymene, α-terpineol, Germacrene D	Adult attractants	*Corylus*	*Phytoptus avellanae*, *Myzocallis coryli*	[91]
	(+/−)-2-hexanol, 3-methyl-3-pentanol, 3,3-dimethyl-1-butanol	Female adult attractants	-	*Aleruodicus dispersus*	[92]
	β-pinene, α-pinene, 1,8-cineole	Antennal active substances	*Quercus palustris*	*Prinobius myardi*	[93]
	(Z) and (E)-β-ocimene, Undecane, Decanal, β-caryophyllene	Female adult attractants	*Salix* spp.	*Nematus oligospilus*	[94]
	Trans-4-thujanol	Adult repellents	*Picea asperata*	*Ips typographus*	[95]
GVs	(S)-linalool, 4,8-dimethyl-1,3,7-nonatriene, (E)-caryophyllene, (R/S)-(E)-nerolidol	Antennal active substances	*Oryza sativa*	*Orseolia oryzivora* Harris and Gagné	[96]
	(Z)-3-hexenol, (Z)-3-hexenyl acetate	Larvae attractants	*Populus nigra*	*Lymantria dispar*	[97]
	Dimethyl disulfide, Geraniol, Eucalyptol, Citronellol	Oviposition repellents	*Broccoli*	*Delia radicum*	[98]
	α-pinene, Trimethylindan, Cyclohexylbenzene	Adult attractants	*Baccharis salicifolia*	*Macrodactylus nigripes*	[99]
	(Z)-3-hexen-1-ol, 3-methyl-butanol, 1-pentanol. Benzaldehyde, 2-hexanone	Female adult attractants	*Prunus armeniaca*	*Capnodis tenebrionis*	[100]
	(E,E)-α-farnesene, (E)-4,8-dimethyl-1,3,7-nonatriene, (E,E)-4,8,12-trimethyl-1,3,7,11-tridecatetraene	Larvae attractants	*Coffea arabica*	*Hypothenemus hampei*	[101]
	Sulcatone	Adult attractants	*Ailanthus altissima*	*Lycorma delicatula*	[102]
	(Z)-3-hexenyl acetate, (E)-β-ocimene	Adult attractants	*Salix babylonica*	*Tuberolachnus salignus*	[103]
	Undecane, Ethyl benzaldehyde, n-hexadecanoic acid	Adult attractants	*Solanum lycopersicum*	*Tuta absoluta*, *Bemisia tabaci*	[104]
	Linalool, Linalyl acetate, Linalool tetrahydride	Antennal active substances	-	*Apolygus lucorum*	[105]
	p-cymene	Adult repellents	*Haplopappusmyiia gregaria*	*Haplopappus foliosus*	[106]
	Methyl salicylate, β-myrcene, (3E,7E)-4,8,12-trimethyltrideca-1,3,7,11-tetraene, α-curcumene	Adult repellents	*solanaceous plants*	*Tuta absoluta*	[107]
	(Z)-3-hexen-1-ol	Oviposition attractants	*Zea mays*	*Ostrinia furnacalis*	[108]
	Z-3-hexenyl acetate	Female adult attractants	*Triticum aestivum*	*Sitodiplosis mosellana*	[109]
GLV	α-pinene, 3-carene, γ-terpinene, Linalool	Adult attractants	*Agave tequilana*	*Scyphophorus Acupunctatus*	[110]
	(E)-2-hexenal, (Z)-3-hexen-1-ol, 1-octen-3-ol, (Z)-3-hexenyl acetate, (Z)-3-hexenyl butyrate	Female adult repellents	*Phaseolus lunatus*	*Spodoptera littoralis Boisd*	[111]
GLVs	(E)-β-caryophyllene	Larvae attractants	*-*	*Nilaparvata lugens*	[112]
	Myrcene, β-phellandrene, Trans-β-ocimene, Terpinolene	Female adult repellents	*Pinus sylvestris*	*Thaumetopoea pityocampa*	[113]
HIPVs	(Z)-3-hexen-1-ol, 4,8-dimethyl-1,3,7-nonatriene, 3-octanone	Cue of host location	*Phaseolus vulgaris*	*Trialeurodes vaporariorum*	[114]
	(E)-8-hydroxy-6-methyl-6-octen-3-one	Aggregation pheromone		*Oulema melanopus*	[115]
	(Germacrene D, β-caryophyllene, Methyl salicylate, E-β-ocimene, (3E)-4,8-dimethyl-1,3, 7-nonatriene	Adult attractants	*Fragaria* × *ananassa*	*Anthonomus rubi*	[116]
	Methyl salicylate, 2-phenylethanol	Adult attractants	*Glycine max*	*Coccinella septempunctata*, *Aphis glycines*, *Chrysoperla carnea and syrphid flies*	[117]
	Methyl salicylate	Adult attractants	*Brassica oleracea*	*Cotesia glomerata, Cotesia rubecula, Pieris caterpillars*	[118]
	Myrcene, (+)-3-carene, β-caryophyllene, (+)-α-pinene, Limonene	Adult attractants	*Pinus armandii*	*Dendroctonus armandi*	[119]
	DMNT	Female adult attractants	*Ulmus minor*	*Oomyzus gallerucae*	[120]
	Methyl jasmonate, Methyl salicylate	Nature enemy attractants	*Zea mays*	*-*	[121]
	1-butyl butylate, 1-hexyl acetate, 1-butyl ethanoate	Adult attractants	*Malus pumila*	*Psyttalia concolor*	[122]
	(Z)-3-hexene-1-ol	Antennal active substances	*Ricinus communis*	*Anomala corpulenta*	[123]
	Linalool, (Z)-3-hexenyl acetate, Geraniol	Male adult attractants	*Glycine max*	*Leguminivora glycinivorella*	[124]
	Linalool, DMNT	Nature enemy attractants	*Zea mays*	*Spodoptera frugiperda*	[125]
	(Z)-3-hexenyl hexanoate, (Z)-3-hexenal	Female adult attractants	*Camellia sinensis*	*Ectropis obliqua*	[126]
HIPVs	Nonanal, (Z)-3 hexenyl acetate, (E)-β-ocimene, (R)-(+)-limonene	Male adult attractants	*Gossypium hirsutum*	*Spodoptera littoralis*	[127]
	(E)-4,8-dimethyl-1,3,7-nonatriene	Nature enemy attractants	*Zea mays*	*Chilo partellus, Trichogramma bournieri, Cotesia sesamiae*	[128]
	α-pinene, β-myrcene, (Z)-3-hexenyl acetate, Limonene, (E)-ocimene, Linalool, DMNT, (E)-β-farnesene, TMTT, Indole	Cue of host location	*Zea mays*	*Polybia fastidiosuscula*	[129]
	DMNT, TMTT	Nature enemy attractants	*Anacardium occidentale*	*Diaphorina citri*	[130]
	Cis-jasmone, Cis-3-hexenyl acetate	Adult attractants	-	*Campoletis chlorideae*	[131]
	β-phellandrene, (E)-β-ocimene, (-)-β-bourbonene, β-ylangene, (E,E)-α-farnesene, Linalool, Myrtenol, (E)(-)-pinocarveol, (E)-pinocamphone, (Z)-pinocamphone, Methyl salicylate	Nature enemy attractants	*Juglans regia*	*Meliboeus ohbayashii*	[132]
	Acetic acid, 2-phenylethanol	Adult attractants	*Malus pumila*	*Aphis pomi, Dysaphis plantaginea*	[133]
	β-ocimene	Female adult repellents	*Camellia sinensis*	*Ectropis obliqua*	[134]
	(Z)-3-hexenal, (E)-2-hexenal, (Z)-3-hexen-1-ol, (E)-2-hexen-1-ol, (Z)-3-hexen-1-yl acetate, 1-hexyl acetate	Female adult attractants	*Triticum aestivum*	*Aelia acuminata*	[135]
	α-pinene, Pseudocumene, Limonene	Nature enemy attractants	*Fragaria* × *ananassa*	*Tetranychus urticae Koch*	[136]

GVs: general volatiles. DMNT: (E)-4,8-dimethyl-1,3,7-nonatriene. TMTT: (E, E)-4,8,12-trimethyltrideca-1,3,7,11-tetraene.

## 5. The Role of VOCs in Multilevel (Aboveground–Belowground) Interactions

In terrestrial ecosystems, the interplay between aboveground and belowground biotic communities exerts a profound influence on ecological functions [137]. While traditional research has concentrated on organisms directly associated with plant roots and shoots, such as root herbivores and foliar herbivores, contemporary studies have broadened to include higher trophic levels, encompassing parasitoids, and non-herbivorous entities like decomposers, symbiotic plant associates, and pollinators. This expanded multitrophic perspective forges connections between the aboveground and belowground food webs, striving for a holistic grasp of the dynamics among plants, herbivores, and carnivores within real-world ecosystems. Plants facilitate interactions within insect communities across the soil–plant boundary via volatile compounds elicited by herbivory, influencing the colonization patterns of various herbivores on host plants [138]. Furthermore, herbivores can modify the emission patterns of plant volatiles, potentially interfering with the host-seeking behaviors of other herbivores and their natural adversaries [139]. In this context, plants serve as conduits for vertical communication, metaphorically described as a “green telephone”, linking subterranean insects with their aboveground counterparts [140]. The onslaught of root-feeding insects prompts the activation of the plant’s stem defense system, culminating in the concentration of toxins within the foliage. This root damage-triggered stem defense reaction can significantly impact the survival, growth, and development of aboveground insectivores. Consequently, a plant-centric functional nexus is forged between subterranean insects and those inhabiting the aboveground realm, weaving a complex tapestry of ecological interactions.

### 5.1. Impact of Aboveground Plant VOCs on Belowground Ecosystem Dynamics

The intricate interplay between aboveground and belowground organisms is a key driver of plant vitality and productivity. These multifaceted interactions are essential for enhancing crop development, optimizing nutrient assimilation, and bolstering defenses against a spectrum of biotic and abiotic challenges [141]. Recent studies have highlighted the significant impact of leaf damage on belowground interactions, suggesting that a decrease in root growth may lead to reduced preferences of subterranean herbivores [142]. The dynamics between surface and soil-dwelling insects, particularly within-species interactions, significantly shape the population structures of rhizosphere and phyllosphere insect communities through the manipulation of host plants [143,144]. These complex interactions also involve higher trophic levels, including parasitic insects [144].

Research on aboveground plant organs reveals that the attack patterns of herbivores—whether singular or multiple—affect the emission rates of VOCs [145]. Plants release specific mixtures of VOCs as multifunctional signals that regulate fungal colonization on plant structures, thereby altering the informational content for aboveground insects [146]. Research has demonstrated that HIPVs play crucial roles in plant defense by attracting natural enemies of herbivores, such as parasitoids and predators, thus providing an indirect defense mechanism [79]. These compounds not only directly impact the feeding behaviors of phytophagous animals [147,148,149] but may also permeate the soil via the root system, partaking in intricate interactions with subterranean microbiota and other organisms [150] (Table 3). For instance, (E)-β-caryophyllene is known to attract entomopathogenic nematodes that parasitize herbivores [151,152]. Additionally, (E)-β-ocimene and other HIPVs can enhance the resistance of neighboring plants by priming them to respond more rapidly and robustly to herbivore attack [17,153,154]. These volatiles can also mediate belowground interactions, influencing root-associated microbial communities and enhancing nutrient uptake [79].

It is noteworthy that VOCs released by aboveground plants, by modulating the physiological and chemical defense mechanisms of plant leaves and roots (Table 3) [148], further influence the interactions between aboveground and belowground plant-herbivore dynamics [150]. Although the role of root-emitted VOCs in inter- and intra-plant signaling is less documented in the scientific literature [155], specific VOCs released by aboveground plants significantly affect the growth and metabolic activities of certain soil microorganisms, underscoring the importance of these compounds in soil ecosystems. These findings not only deepen our understanding of the chemical communication networks within plant ecosystems but also pave the way for new research directions in exploring the interactions of VOCs within ecosystems (Figure 3).

### 5.2. The Impact of Belowground Biodiversity on Aboveground Plant Performance

Belowground communities typically harbor greater biodiversity than their aboveground counterparts, yet the factors governing this diversity remain incompletely understood [156]. Plants orchestrate interactions between aboveground and belowground herbivores through alterations in growth, nutrition, and defense metabolites [157], thereby influencing herbivore populations in aspects such as host selection, survival, growth, and reproduction [158,159]. Root infections commonly impair the performance of foliar herbivores and influence the oviposition decisions of aboveground insects [160]. Feeding by subterranean herbivores may lead to systemic changes in the defense levels of plant stems, impacting both aboveground herbivores and organisms at higher trophic levels [161].

Plants indirectly regulate interactions between root and stem herbivores through alterations in primary and secondary metabolites (Table 3) [2]. This effect can cascade to affect organisms at higher trophic levels (Table 3). Recent studies have shown that belowground phytophagous insects can disrupt the host location behaviors of aboveground parasitoids, thus affecting plants’ indirect defenses. Belowground herbivory influences aboveground herbivore performance and feeding preferences through plant chemical changes [145]. The decomposition of insect exuviae participates in multitrophic interactions mediated by aboveground volatiles [1].

Plants establish multitrophic interactions with arthropods and microbes both above and below the ground [162]. Trichoderma, a common soil fungus, colonizes roots and promotes plant growth and health, and it has been shown to induce plant resistance against foliar herbivorous insects [163]. The productivity of aboveground plants is strongly influenced by belowground microbes, including pathogenic microbes with negative effects, and beneficial microbes, such as nitrogen-fixing bacteria and mycorrhizal fungi (Table 3). The belowground interactions between plants and functionally diverse microbial groups extend to aboveground plant companions, such as herbivores and their natural enemies [164].

**Table 3 insects-15-00572-t003:** The research of multilevel (aboveground–belowground) interactions.

Belowground–Plants–Aboveground	Multilevel (Aboveground–Belowground) Interactions	Reference
*Delia radicum*/*Pieris brassicae*–*Brassica rapa*	Shoots of doubly-infested plants emitted higher levels of methanol.	[145]
*Pieris brassicae*–Brassicaceous plants–*Trybliographa rapae*	Simultaneous attacks by BG and AG herbivores result in lower levels of parasitism in the field.	[162]
*Populus nigra*–*Melolontha*	BG herbivory in poplar can alter the feeding preference of AG herbivores through phytochemical changes triggered by root-to-shoot signaling.	[149]
*Arbuscular mycorrhizal*–Aphid	AM fungi play a crucial bottom-up role in insect host location by enhancing the attractiveness of plant VOCs to aphids, while aphids, in turn, inhibit the formation of AM symbioses	[165]
*Candidatus Liberibacter asiaticus*–*Steinernema diaprepesi*–*Diaprepes abbreviatus*	Nonfeeding larvae associated with CLas-infected plants were equally susceptible to infection by entomopathogenic nematodes (EPNs) as those near noninfected plants experiencing larval damage from *Diaprepes abbreviatus*.	[166]
*Spodoptera frugiperda*–*Acalymma vittatum*–*Cucumis sativus*	Decreased preference by belowground herbivores in this system may be due to reduced root growth.	[142]
*Acalymma vittatum* larvae–*Acalymma vittatum* adults	β-ocimene alone can elicit plant resistance against squash bugs.	[17]
*Acalymma vittatum* larvae–*Cucumis sativus*	Despite the positive effect of belowground damage on honeybee visitation, root herbivory had a stronger negative impact on plant reproduction compared to leaf herbivory.	[148]
Insect exuviae–*Diadegma semiclausum* and *Diaeretiella rapae*	Soil amendment with *Acheta domesticus* or *Tenebrio molitor* exuviae resulted in the increased attraction of the two parasitoid species.	[1]
*Trichoderma harzianum*–*Phyllophaga vetula*	*T. harzianum* stimulated the emission of sesquiterpenes like β-caryophyllene and δ-cadinene.	[6]
*Bikasha collaris*–*Spodoptera litura*	Interactions between conspecific herbivores in aboveground and belowground environments could influence plant rhizosphere and phyllosphere herbivore communities by altering volatiles emitted from both aboveground and belowground sources.	[139]
*Meloidogyne arenaria*–*Nicotiana tabacum*–*Myzus persicae*	Tobacco plants detect the presence of EPNs in the rhizosphere, leading to modifications in their interactions with both aboveground and belowground herbivores.	[2]
*Brassica nigra*–*Delia radicum*–*Cotesia glomerata*	The foraging behavior of a parasitoid that targets aboveground herbivores can be influenced by belowground herbivores due to changes in the plant’s volatile blend.	[167]
*Agriotes lineatus*–*Spodoptera littoralis*	Belowground and aboveground herbivores exhibit indirect interactions that influence behavioral avoidance strategies, affecting both oviposition and larval feeding decisions.	[147]
*Diabrotica virgifera virgifera*–*Zea mays*–*Spodoptera littoralis*	The parasitoid demonstrated the ability to learn and distinguish between different odor emissions, enhancing its response to the odor emitted by a plant infested by both aboveground and belowground herbivores after encountering this odor in the presence of hosts.	[143]
*Heterodera schachtii*–*Arabidopsis thaliana*–*Frankliniella occidentalis*/*Tetranychus urticae*	While thrips actively avoid plants infected with nematodes, spider mites exhibit a preference for these plants. Moreover, nematode infection significantly improves the life-history performance of *Tetranychus urticae*.	[168]
*Trichoderma longibrachiatum*–*Solanum lycopersicum*–*Macrosiphum euphorbiae*	Compared to uncolonized controls, plants with roots colonized by *T. longibrachiatum* MK1 exhibited quantifiable differences in specific VOC emissions, enhanced aphid population growth indices, increased attractiveness to aphid parasitoids and predators, and accelerated development of aphid predators.	[169]
*Acalymma vittatum*–*Zucchini squash*–*Anasa tristis*	Belowground herbivory enhances aboveground plant resistance and deters aboveground foraging herbivores. Additionally, plants damaged by larvae emitted higher amounts of a key volatile compound, (E)-β-ocimene, compared to non-damaged controls.	[153]
Western corn rootworm–*Zea mays*–European corn borer	Undecane, which is absent from the volatile bouquet of healthy plants, was the sole compound upregulated upon root infestation and acted as a repellent against initial oviposition.	[160]
*Meloidogyne incognita*–*Sitobion avenae*	*M. incognita* infection-induced changes in jasmonic acid, salicylic acid, and volatile content, which mediated the plant’s response to *S. avenae*.	[170]
*2,3-butanediol*–*Zea mays*–*Cotesia marginiventris*	The application of 2,3-BD to the headspace of the plants did not affect the parasitoids, whereas application to the soil increased parasitoid attraction.	[171]
*Azospirillum brasilense*–*Diabrotica speciosa*	As inoculation by *A. brasilense* induces higher emissions of (E)-β-caryophyllene compared with non-inoculated plants, the non-preference of *D. speciosa* for inoculated plants is likely related to this sesquiterpene.	[172]
*Globodera pallida*–*Solanum tuberosum*–*Myzus persicae*	The systemic response of the potato plant following infection with *G. pallida* indirectly influences the performance of *M. persicae*.	[173]

AG: aboveground. BG: belowground. EPNs: entomopathogenic nematodes.

Plants mediate interactions within insect communities across the soil–plant interface through HIPVs, thereby affecting the colonization of different herbivores on host plants. The interactions between aboveground and belowground conspecific herbivores shape plant rhizosphere and phyllosphere herbivore communities by altering the emission of aboveground and belowground volatiles and regulating the colonization of other herbivores and associated natural enemies [139]. Multitrophic plant-insect interactions are mediated by plant volatiles. The emission of herbivore-induced plant volatiles is influenced by environmental conditions, such as soil microbes and nutrients, and impacts aboveground trophic interactions. VOCs released by soil microbes affect plant growth and resistance to pathogens. Plants convey ecologically relevant information to neighboring plants through chemical signals, such as the production of HIPVs like (E)-β-ocimene [153], triggered by insect herbivory, which can enhance the defensive capabilities of neighboring plants. HIPVs are released from both directly damaged plant tissues and systemically from undamaged tissues. β-Ocimene alone can induce plant resistance to *Acalymma vittatum* [17]. The release of aboveground HIPVs recruits natural enemies of herbivores as an indirect induced plant defense. After spider mite attack, the mycorrhizae of *Phaseolus vulgaris* significantly alter the composition of HIPVs, increasing the release of β-ocimene and β-caryophyllene [174].

## 6. Future Research Directions and Applications

### 6.1. Addressing Gaps and Forging New Paths in VOCs Research

Contemporary research on ecosystem interactions often centers on isolated systems or a limited selection of plant species, which fails to provide comprehensive validation across varied ecosystems and a multitude of plant varieties. This constrained focus is exacerbated by the dominance of short-term experiments that do not encapsulate the complex and persistent interactions inherent in natural ecosystems. The emission and subsequent impact of VOCs are subject to significant variability, influenced by biodiversity and environmental factors such as temperature, humidity, and CO_2_ levels Despite their critical importance, these variables are frequently neglected or inadequately considered in current studies. To address this deficiency, there is an urgent need for comparative studies of VOCs spanning diverse ecosystems, including forests, grasslands, and wetlands. Such investigations would shed light on the influence of different plant species and overall species richness on VOCs’ emissions and responses. Furthermore, long-term observational studies in natural settings are vital to deepen our understanding of VOCs’ dynamics and their role in sustaining ecosystem equilibrium. Recent systematic reviews have underscored the chemical diversity of invasive species and their modified VOCs’ emissions in native versus invaded territories, indicating that these emissions are inheritable traits that contribute to invasiveness [175]. Additionally, the role of VOCs derived from soilborne pathogenic and beneficial fungi, particularly mycorrhizae, in influencing plant performance has been emphasized, showing how plant VOCs regulate plant–fungi interactions [5].

Ultimately, comprehensive research designs that incorporate the effects of biodiversity and environmental factors on VOCs’ release and ecological interactions will yield more holistic and accurate insights, reflecting the true complexity of our natural world. Incorporating the latest findings, such as the influence of invasive plants and soilborne fungi on VOCs’ emissions, will enhance our understanding of these complex ecological processes [5,175].

### 6.2. Harnessing Plant VOCs: From Defense Mechanisms to Innovative Pest Control Strategies

Plant-derived VOCs play a crucial role in the interactions between plants, insects, and herbivores. Individual VOCs can exhibit toxicity or deterrence, function as signaling molecules to attract natural enemies, or be perceived by distant plant tissues as signals to initiate defense responses [176]. During invasions by alien species, VOCs facilitate interactions between native and introduced plants and insects, influencing communication among native flora and altering the host-seeking behavior of local insects [177].

Studies have demonstrated that spraying plants with essential oils can trigger plant defenses against pests. Utilizing different cruciferous plants can redistribute cabbage root flies in broccoli crops without compromising the control exerted by natural enemies. The importance of plant volatiles in pest management points to the potential development of a semiochemical-assisted “push–pull” system, where trap plants increase efficacy by synthesizing and emitting attractive compounds [178,179]. Integrating pheromones with plant attractants has proven more effective than using plant attractants alone, suggesting this combination as a potentially superior field application method [180].

For instance, merging nectar rewards from plants with HIPVs such as methyl salicylate (MeSA) and methyl anthranilate (MeA) in an “attract and reward” strategy can entice natural enemies into treated crops, providing sustenance and refuge. This approach maximizes their survival and residency, enhances parasitism rates, and diminishes pest populations [181]. Ongoing research suggests that chloroplasts are optimal sites for the metabolic engineering of EF synthase genes from plant sources, potentially leading to the creation of transgenic plants capable of synthesizing and emitting alarm pheromones to deter aphids [182].

## 7. Conclusions and Prospects

In this review, we delved into the role of VOCs in multitrophic interactions and their crucial function in the intricate communication network between plants, insects, and antagonists. The diversity of VOCs and their dual role in plant defense mechanisms—encompassing both direct and indirect defenses—highlight their importance within ecosystems. By modulating chemical communication between plants and insects, VOCs are vital for preserving ecological equilibrium and fostering biodiversity.

We also examined the role of VOCs in multilevel interactions, demonstrating how they influence plant growth and performance, as well as their capacity to regulate interactions between plants and microbial communities. These insights underscore the potential of VOCs in enhancing agricultural productivity and ecosystem health, particularly in the development of innovative pest control strategies and the improvement of crop resilience.

Recent advancements in VOCs’ research have introduced rapid methods for identifying and quantifying VOCs’ profiles, which can significantly improve our ability to monitor and manipulate VOCs-mediated interactions. Techniques such as gas chromatography–mass spectrometry and proton transfer reaction–mass spectrometry offer precise and efficient ways to analyze VOCs. Researchers can leverage these methods to develop targeted pest management strategies, optimize plant breeding programs for enhanced VOCs’ production, and monitor ecosystem health more effectively.

Future research should focus on comparative studies across diverse ecosystems and the significance of long-term observational research to better understand VOCs’ dynamics. For instance, recent studies have demonstrated that VOCs’ emissions can vary significantly between different plant species and environmental conditions, providing new insights into how VOCs’ function in various ecological contexts. These studies promise a more comprehensive grasp of the complexity of VOCs in the natural environment and bolster the sustainable advancement of agriculture and ecosystem management.

In summary, the multifaceted functionality of VOCs in plant defense and ecosystem processes, coupled with their prospects in future agricultural and ecological research, converge on a singular insight: VOCs are an essential component of the natural world. Their continued study is poised to uncover latent connections and novel applications within the tapestry of life. As we deepen our knowledge of these complex interactions, research into VOCs will equip us with innovative means to safeguard and nurture our invaluable ecosystems.

## Figures and Tables

**Figure 1 insects-15-00572-f001:**
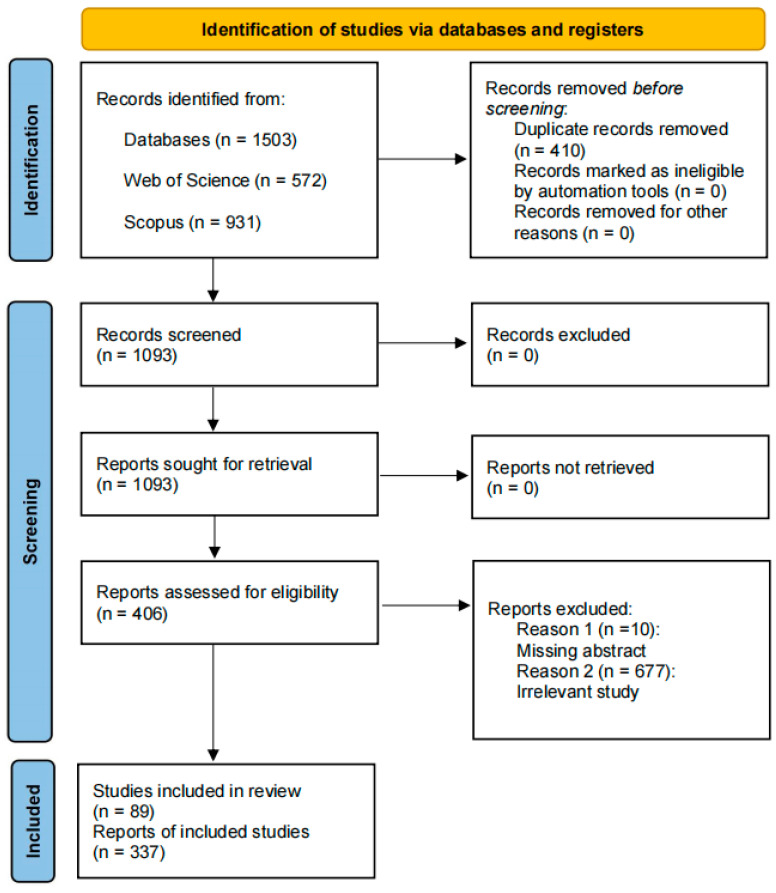
Flowchart research methodology for databases extraction.

**Figure 2 insects-15-00572-f002:**
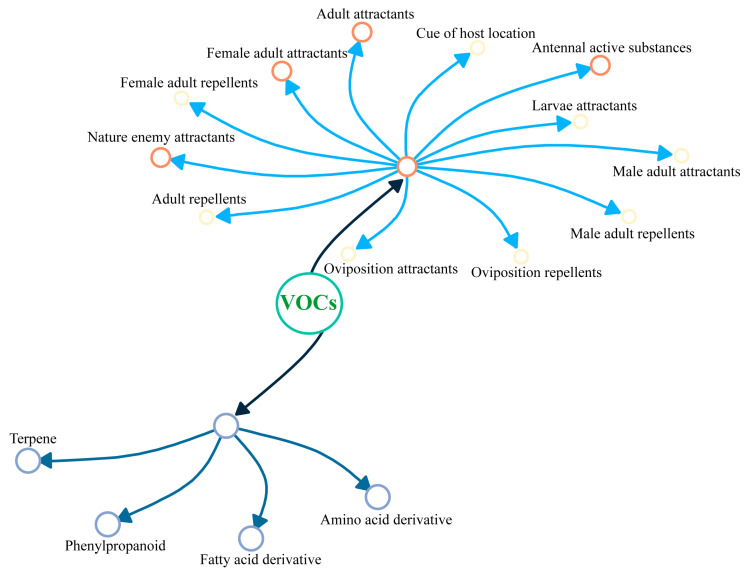
Types and functions of VOCs. Node sizes represent the frequency of related functional research in the network visualization.

**Figure 3 insects-15-00572-f003:**
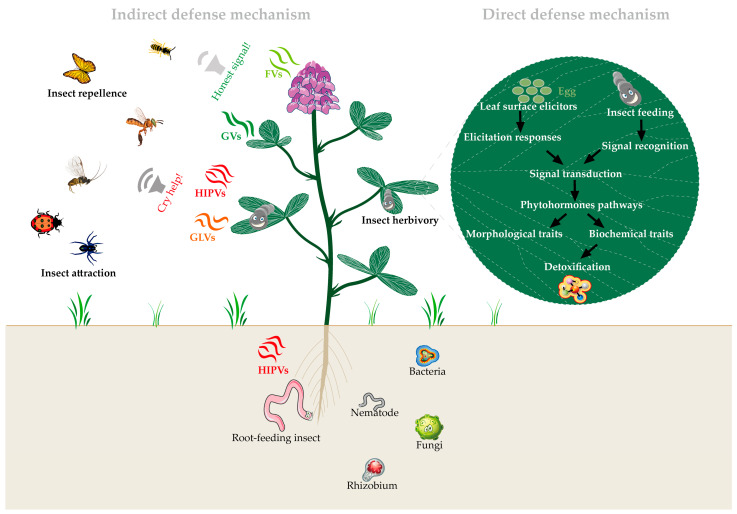
A network of plant defenses against insect herbivory. GVs: general volatiles. FVs: floral volatiles. HIPVs: herbivore-induced plant volatiles. GLVs: green leaf volatiles.

**Table 1 insects-15-00572-t001:** PRISMA flowchart for data extraction in systematic reviews.

Database Query String	Search String Strategy Boolean Operators	No. Articles	No. Deduplicated Articles
Web of Science	TS 1 = (“Volatile Organic Compounds” OR VOCs OR “volatile compounds” OR “plant volatiles” OR “organic volatiles”) AND (TS = (“multitrophic interactions” OR multitrophic OR “trophic levels” OR “plant–insect interactions” OR “plant–pest interactions” OR “plant–antagonist interactions” OR “plant–herbivore interactions” OR “plant-defense mechanisms”))	463	1093
	TS 2 = (“Volatile Organic Compounds” OR VOCs OR “volatile compounds” OR “plant volatiles” OR “organic volatiles”) AND (TS = (“aboveground–belowground interactions” OR “belowground interactions” OR “aboveground interactions” OR “soil–plant interactions” OR “root–shoot interactions”))	89	
	TS 3 = (“multitrophic interactions” OR multitrophic OR “trophic levels” OR “plant–insect interactions” OR “plant–pest interactions” OR “plant–antagonist interactions” OR “plant–herbivore interactions” OR “plant-defense mechanisms”) AND (TS = (“aboveground–belowground interactions” OR “belowground interactions” OR “aboveground interactions” OR “soil–plant interactions” OR “root–shoot interactions”))	20	
Scopus	TITLE-ABS-KEY1: ((“Volatile Organic Compounds OR vocs OR “volatile compounds” OR “plant volatiles” OR “organic volatiles” OR “volatile secondary metabolites”) AND (“multitrophic interactions” OR multitrophic OR “trophic levels” OR “plant–insect interactions” OR “plant–pest interactions” OR “plant–antagonist interactions” OR “plant–herbivore interactions” OR “plant-defense mechanisms” OR “plant–biotic interactions”))	908	

## Data Availability

This study did not create or analyze new data; therefore, data sharing is not applicable to this article.

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
