# Peer review of "Multitrophic and Multilevel Interactions Mediated by Volatile Organic Compounds"

_insects, 2024, doi:10.3390/insects15080572_

Round 1

Reviewer 1 Report

Comments and Suggestions for Authors

The review entitled “Multitrophic and multilevel interactions mediated by Volatile Organic Compounds offers valuable information about the roles of VOCs in the mechanisms of plant defenses and their influence in multitrophic interactions. It is a comprehensive and detailed work, that synthesizes a wide range of information in a single manuscript. For that reason, I would like to congratulate the authors. Consequently, in my opinion, the manuscript can be accepted for publication in Insects Journal although but after addressing these details:

-It would be preferable if all figures and tables were not interrupted by text and always appear after the paragraph in which they were first cited.

-Figure 2: There is a lack of one word in the scheme.

-In “6.2. Harnessing plant VOCs: From defense mechanisms to innovative pest control strategies” I personally thing that this section could integrate the following work that is focused in the development of HIPVs as defense elicitors:

Pérez-Hedo, M., Alonso-Valiente, M., Vacas, S., Gallego, C., Rambla, J. L., Navarro-Llopis, V., ... & Urbaneja, A. (2021). Eliciting tomato plant defenses by exposure to herbivore induced plant volatiles. Entomologia Generalis, 41(3), 209-218.

Author Response

Dear reviewer,

Reviewer 2 Report

Comments and Suggestions for Authors

I suggest the revision of minor details included in the manuscript

Author Response

Dear reviewer,

Reviewer 3 Report

Comments and Suggestions for Authors

Dear Authors 

I have attached my comments 

Regards

Author Response

Dear reviewer,
